# Estimation of Rubber Yield Using Sentinel-2 Satellite Data

**Niwat Bhumiphan** [1], **Jurawan Nontapon** [1], **Siwa Kaewplang** [1,*], **Neti Srihanu** [2], **Werapong Koedsin** [3]
**and Alfredo Huete** [3,4]

1   Faculty of Engineering, Mahasarakham University, Kantharawichai District, Maha Sarakham 44150, Thailand
2   Faculty of Engineering, Northeastern University, Muang District, Khon Kaen 40000, Thailand
3   Faculty of Technology and Environment, Phuket Campus, Prince of Songkla University, Phuket 83120, Thailand
4   School of Life Sciences, University of Technology Sydney, Sydney, NSW 2007, Australia
*   Correspondence: siwa.kae@msu.ac.th; Tel.: +66-81-547-5825

**Abstract:** Rubber is a perennial plant grown to produce natural rubber. It is a raw material for industrial and non-industrial products important to the world economy. The sustainability of natural rubber production is, therefore, critical for smallholder livelihoods and economic development. To maintain price stability, it is important to estimate the yields in advance. Remote sensing technology can effectively provide large-scale spatial data; however, productivity estimates need to be processed from high spatial resolution data generated from satellites with high accuracy and reliability, especially for smallholder livelihood areas where smaller plots contrast with large farms. This study used reflectance data from Sentinel-2 satellite imagery acquired for the 12 months between December 2020 and November 2021. The imagery included 213 plots where data on rubber production in smallholder agriculture were collected. Six vegetation indices (Vis), namely Green Soil Adjusted Vegetation Index (GSAVI), Modified Simple Ratio (MSR), Normalized Burn Ratio (NBR), Normalized Difference Vegetation Index (NDVI), Normalized Green (NR), and Ratio Vegetation Index (RVI) were used to estimate the rubber yield. The study found that the red edge spectral band (band 5) provided the best prediction with $R^2 = 0.79$ and RMSE = 29.63 kg/ha, outperforming all other spectral bands and VIs. The MSR index provided the highest coefficient of determination, with $R^2 = 0.62$ and RMSE = 39.25 kg/ha. When the red edge reflectance was combined with the best VI, MSR, the prediction model only slightly improved, with a coefficient determination of ($R^2$) of 0.80 and an RMSE of 29.42 kg/ha. The results demonstrated that the Sentinel-2 data are suitable for rubber yield prediction for smallholder farmers. The findings of this study can be used as a guideline to apply in other countries or areas. Future studies will require the use of reflectance and vegetation indices derived from satellite data in combination with meteorological data, as well as the application of complex models, such as machine learning and deep learning.

**Keywords:** natural rubber; smallholder; Sentinel-2; yield estimation model; reflectance

## 1. Introduction

Rubber (*Hevea brasiliensis*) is a perennial plant grown to produce natural rubber. It also helps to absorb carbon dioxide stored in the form of biomass [1]. There are over 11 million hectares of agricultural land worldwide, with rubber plantations covering about 9.2 million hectares (78%) in Southeast Asia, about 3.67 million hectares (31%) in Indonesia, and about 3.23 million hectares in Thailand [2]. Most rubber trees are grown in the tropics and have an economic life of 30 to 35 years [3]. The rubber trees are cut down to provide timber for trade; every year, about 3–4% of the rubber plantation area is cleared and replaced by a new generation of rubber [4,5]. Last century, rubber trees were planted over large areas. Currently, rubber cultivation is gradually shifting to the smallholder sector [6]. Most Southeast Asian countries converted to smallholder agriculture in 2018, and they contribute more than three-quarters of the world's natural rubber production, with the main natural rubber producing countries being Thailand, Indonesia, Vietnam, India, China,

and Malaysia. In 2018, natural rubber production reached 14.33 million tons [2]. Thailand, of course, is located in the tropics, where the environment is suitable for the cultivation of rubber. In the past, rubber cultivation in Thailand was popular in the southern and eastern regions; however, the expansion rate of rubber plantations in the southern region has begun to decline. With policies aimed at reducing the amount of land unsuitable for planting rubber trees and allowing farmers to plant oil palms or fruit trees instead, rubber plantations have recently expanded to the northeast, where they had never been planted before [7]. It is obvious that rubber is a significant commercial crop for Thailand.

Natural rubber production is vital to more than 20 million farmers worldwide. Over the last century, rubber cultivation has gradually shifted to the smallholder [8], and this sector accounts for more than 75% of the world's natural rubber production [9,10]. The size of rubber plantations is assessed differently by each country; however, it is reported that most smallholder farmers have less than 4 hectares of land [11,12]. Smallholders in Malaysia, Thailand, Myanmar, India, and Indonesia produced natural rubber at a high rate (i.e., between 85% and 93%) [13]. However, smallholder rubber farmers continue to have limited income and limited access to financial resources [14,15]. Therefore, creating sustainability in natural rubber production, especially for smallholders, is crucial for developing the global economy.

Remote sensing technology can effectively provide large spatial data [16–20]. Satellite imagery sources are often freely available; they cover a large geographic area and have high temporal resolution. Since satellite images have multiple spectra, different photo indices are presented to distinguish harvesting areas [21], and they play an important role in mapping local and regional rubber plantations [22–24]. Furthermore, these tools have provided a real-time understanding of changes in rubber plantation area [25,26]. This will reduce the time, labor, and cost of inspecting rubber plantations in the field, where access is limited [27,28]. Because estimating yields is important to maintaining price stabilization, many countries use common techniques to collect yield data and area-based field reports. Most of these data come from post-harvest surveys conducted relatively late [29]. Satellite remote sensing is used for various problems and applications [30], including yield estimation and agricultural yield prediction. It is an interesting research work and of great importance [28]. Agricultural yield assessment and forecasting comprise an essential component [1,31]. Several studies have utilized vegetation indices to estimate the yield of crops such as rice, wheat, barley [32–36], potato [37–39], maize [40–44], and oil palm [45–49]. Obviously, the application of remote sensing in agriculture, including yield estimation and crop forecasting, is of great research interest and importance [34,38,50,51]. Several studies utilized low-resolution MODIS satellites to solve the classification problem based on biophysical properties from time series, which provides good performance [23,34,52]. The Landsat satellites have a moderate spatial resolution (i.e., 30 m spatial resolution) and have been accurately used for estimating rubber tree growth and age [53–56]. The THEOS satellite, in particular, has been used to analyze the change and expansion of the rubber tree area and the rate of change in northeast Thailand [7]. However, most studies utilized low-resolution (i.e., MODIS), and a study used moderate-resolution but low temporal resolution (i.e., THEOS) satellites for rubber yield applications, which is not reasonable for smallholder farms with small plot sizes (less than 4 hectares of land). It is found that satellite data are usually used for applications in the rubber industry; however, the use of satellite imagery to estimate rubber yield has not been very effective, and the satellites used have low spatial resolution [57]. The Sentinel-2 satellites have a high resolution, allowing them to distinguish between rubber plantations and complex natural forests [58]. Since the Sentinel-2A and Sentinel-2B satellites work together, they have high temporal resolution and optimum spatial resolution, allowing the utilization of time series data [59], and they may be suitable for tasks that require continuous monitoring and small areas. Various research has focused on increasing the production of natural rubber in wide areas but has not considered the constraints of smallholder primary producers. In addition, it has not been clearly stated how the information on rubber yields for smallholders will be

collected [60]. Therefore, this study aims to test the Sentinel-2 satellite in the application of rubber yield estimation for smallholder farms. Both spectral and vegetation indices from Sentinel-2 were used as inputs to the models. Linear and multiple linear regressions were used to build prediction models. Each model was statistically compared to determine the most suitable model for rubber monthly yield prediction. The accurate model will be helpful to stabilize the country's rubber prices in the future.

## 2. Materials and Methods

### 2.1. Study Area

This study was conducted in Bueng Kan, Thailand, a province with a perfect environment, which has the ninth largest amount of area devoted to rubber plantations in the country and the largest rubber plantation in the Northeast. The province is located in the upper northeast of Thailand at 17°46′ to 18°26′ N and 103°14′ to 104°11′ E and covers a lowland area of about 4300 km$^2$ (Figure 1). The average elevation in the region is 200 m above mean sea level, and the weather is favorable, with an average temperature of 27 °C and average rainfall between 1500–2500 mm per year.

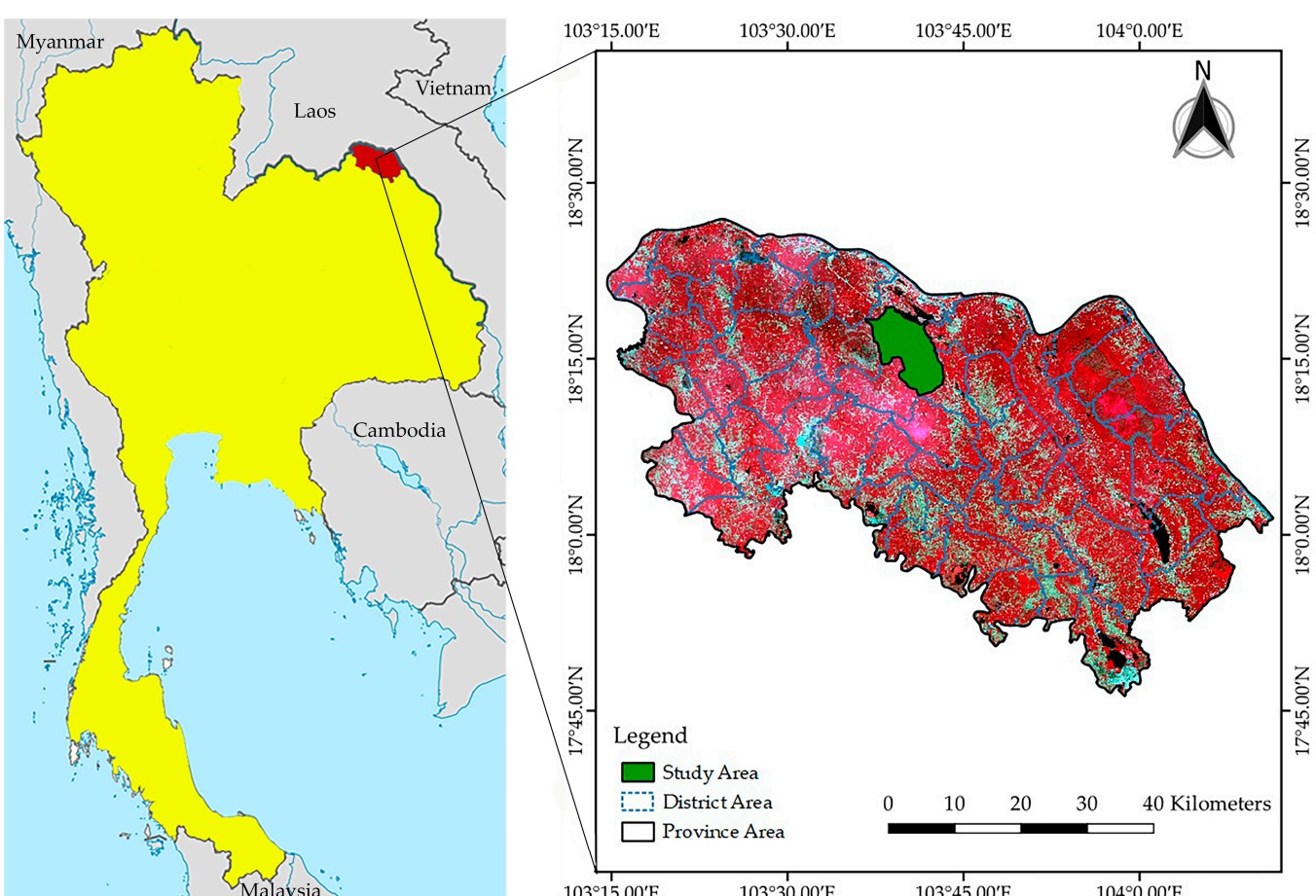

**Figure 1.** Geographic location of the study area in the image on the right is the Sentinel-2 satellite, the Non Somboon Subdistrict, Muang District, Bueng Kan Province, Thailand.

The main crops are rubber, rice, and oil palm, with rubber plantations occupying the largest area with 775,994 rai (124,159.04 hectares), accounting for 59.90%; in-season rice 492,181 rai (78,748.96 hectares), accounting for 38%; oil palm 14,184 rai (2269.44 hectares), accounting for 1.09%; and off-season rice 13,030 rai (2084.8 hectares), accounting for 1.01% of the total cultivated area, (Office of Agricultural Economics 2014). The Rubber Research Institute Department of Agriculture (1993) introduced rubber varieties for farmers in the

Northeast, where the typical plantations are between 5 and 30 rai in area, and most of the rubber plant species planted in the study area were RRIM 600.

### 2.2. Datasets and Processing

### 2.2.1. Field Survey Data

The field data collection was conducted from December 2020 to November 2021. The selected experimental plots are located in the Non Somboon Subdistrict, Muang District, Bueng Kan Province. Field data were gathered by interviewing 213 rubber farmers about the varieties planted, planting year, and rubber harvest year (Figure 2). Selected plots were required to have an area of at least 5 rai (0.8 hectares) per plot and an age of between 5 and 25 years. The farmers' sales invoices providing weight data (number of kilos) were used together with consultation with the relevant authorities in the study area to verify the actual field data. Rubber yields from rubber plantations are sold twice a month.

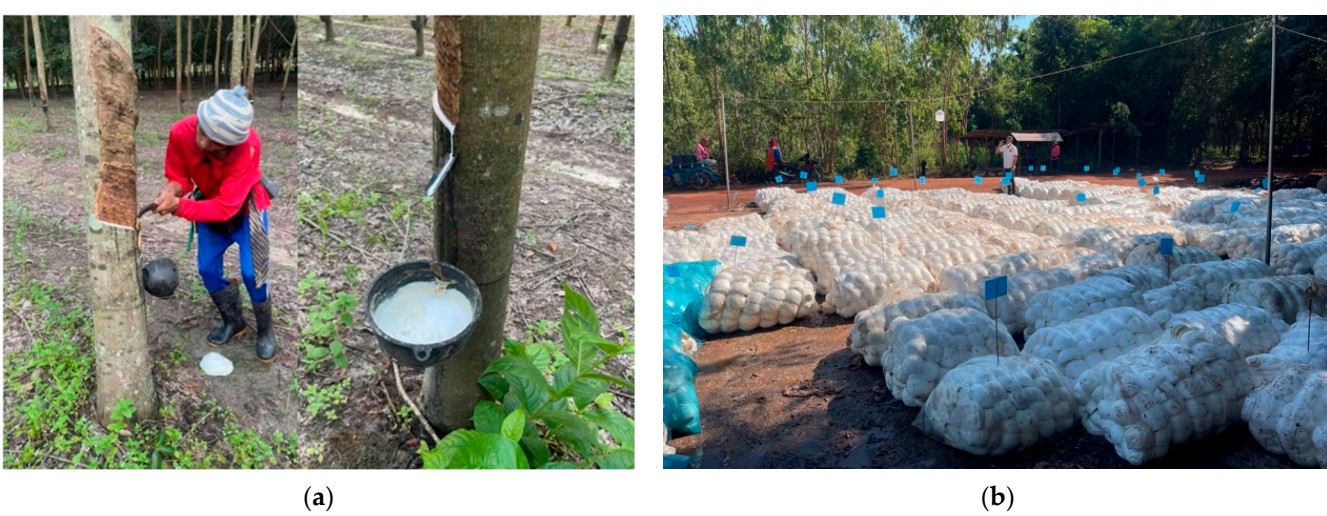

(**a**)  (**b**)

**Figure 2.** Shows the tire yield data collection. (**a**) Conducted interviews with smallholder rubber farmers; (**b**) Interviewed staff. The data were used to verify the field data.

### 2.2.2. Satellite Images

Sentinel-2 satellite data that covered the study area (213 plots) from December 2020 to November 2021 (12 months) were downloaded from the USGS Earth Explorer (USGS Earth Explorer. Available at: https://earthexplorer.usgs.gov/, accessed on 14 December 2020), a free resource for satellite imagery. Then, radio-metric correction and geometric correction were performed [61]. The 20-m spatial resolution bands were re-sampled to 10 m. In this study, 40 ground control points were used for geometric correction. The rubber cultivation area was classified by a supervised classification technique using a semi-automatic classification plugin in the QGIS program. Six vegetation indices, namely Green Soil Adjusted Vegetation Index (GSAVI), Modified Simple Ratio (MSR), Normalized Burn Ratio (NBR), Normalized Difference Vegetation Index (NDVI), Normalized Green (NR), and Ratio Vegetation Index (RVI), were used to determine the characteristics of the rubber trees, and to classify rubber plantations [58]. The vegetation indices were calculated using the following equation.

$$GSAVI = \frac{RED - GREEN}{RED + GREEN + 0.5} \times (1 + 0.5) \tag{1}$$

$$MSR = \frac{RED}{(NIR/RED + 1)^{0.5}} \tag{2}$$

$$NBR = \frac{(NIR - SWIR2)}{(NIR + SWIR2)} \tag{3}$$

$$NDVI = \frac{(NIR - RED)}{(NIR + RED)} \tag{4}$$

$$NR = \frac{RED}{(NIR + RED + GREEN)} \tag{5}$$

$$RVI = \frac{RED}{NIR} \tag{6}$$

where RED is the visible wavelength of red light, GREEN is the visible wavelength of green light, NIR is the near-infrared wavelength, and SWIR2 is the shortwave infrared-2 wavelength.

### 2.2.3. Data Processing and Data Analysis

To select suitable features (i.e., bands and vegetation indices), a calculation of the correlation coefficient between monthly yield and spectral features was carried out [62]. The selected features were then used to build linear and multiple linear regressions to estimate the monthly rubber yield. The field data were separated into two parts, training and testing, which were used to train and evaluate the model, respectively. To prevent bias, the data were rotated 50 times, resulting in 50 differences between the training and testing data sets. The decision coefficient ($R^2$) was calculated and reported. The significance level at 5% ($p \leq 0.05$) was set (Figure 3).

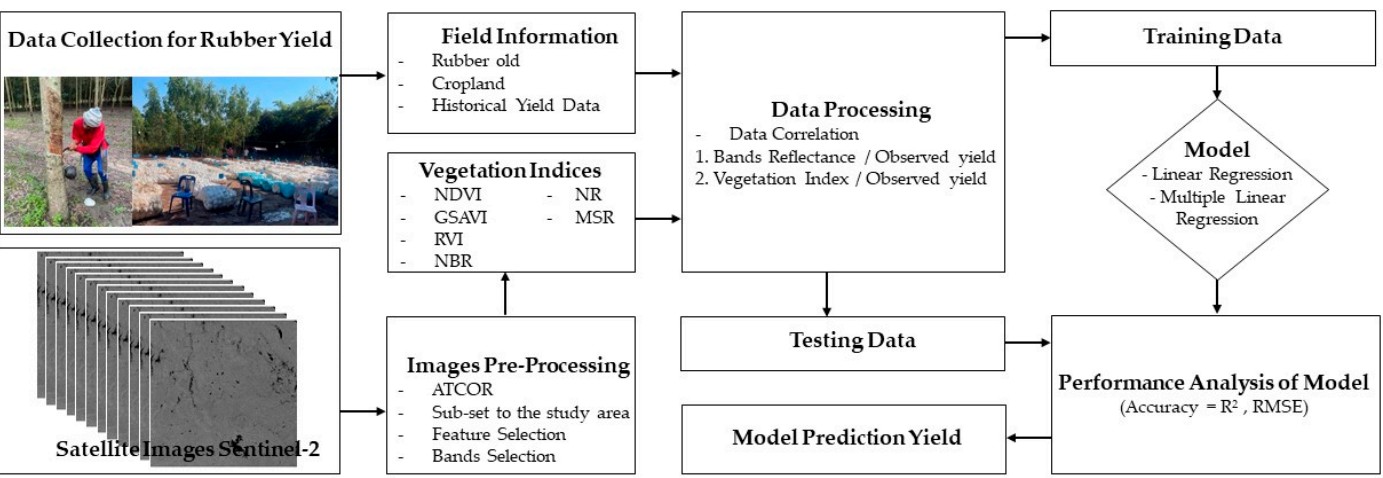

**Figure 3.** Yield prediction generation and analysis.

## 3. Results and Discussion

### 3.1. Data Correlation

The correlations between bands (reflectance) and vegetation indices with rubber yield are reported in Figures 4 and 5, respectively. The correlation coefficient (R) is displayed as a Correlation Heatmap where R is without unit and ranges from −1 to 1. R values close to −1 indicate that the variables are negatively and strongly correlated. R values close to 1 indicate that the variables are positively and strongly correlated, while an R value close to 0 indicates that the variables are weakly or non-linearly correlated. For the spectral reflection of 13 bands, it was found that the best correlations were B4 (Red), B5 (vegetation red edge), B11 (SWIR), and B12 (SWIR) [63]. The best correlation of spectral reflectance ranged from 0.71–0.89, with statistical differences at a significant level of ≤0.05, while the best correlation of vegetation indices with rubber yield were NBR, NR, and MSR. The

best correlation of vegetation indices ranged from 0.74–0.80, with a statistically significant difference of ≤0.05.

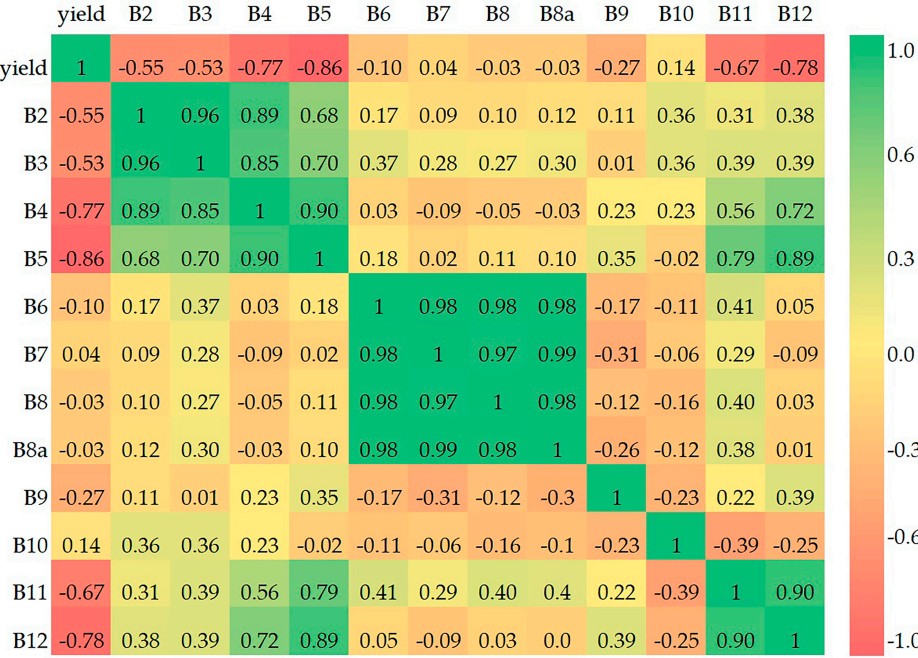

**Figure 4.** The correlation heatmap of band reflectance and observed yield. The color value of the right stripe from high to low shows the strength of the factor correlation.

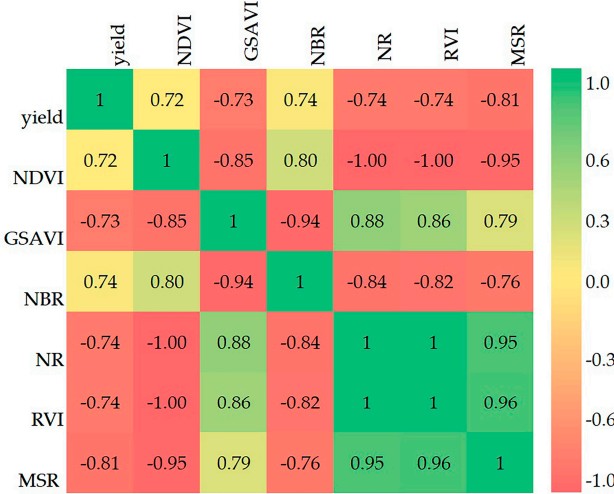

**Figure 5.** The correlation heatmap of vegetation indices and observed yield. The color value of the right stripe from high to low shows the strength of the factor correlation.

### 3.2. Rubber Yield Prediction Models

#### 3.2.1. Linear Regression Analysis

Suitable spectral reflectance and the six vegetation indices were used to build a linear regression model to predict the rubber yield. The results showed that the coefficient of determination ($R^2$) of the models obtained from the spectral reflectance ranged from 0.47–0.79, and the RMSE ranged from 29.63–45.25 kg/ha (Table 1). The best model was selected, and a scatter plot was created to graph the relationships between the observed yields and predicted yields (Figure 6) by setting the rubber yield based on the observed yield as the dependent variable (Y) and the predicted yield as the independent variable (X).

The model derived from vegetation indices showed that the coefficient of determination ($R^2$) ranged from 0.52–0.62, and the RMSE ranged from 39.25–44.81 kg/ha (Table 2). The best model was selected, and a scatter plot was created to graph the relationships between the observed yields and predicted yields (Figure 7) by setting the rubber yield based on the observed yield as the dependent variable (Y) and the predicted yield as the independent variable (X).

**Table 1.** The $R^2$ and RMSE of linear regression models of the reflectance of the Sentinel-2 satellite.

| Bands Reflectance | Training Model | | Testing Model | | Equation |
|---|---|---|---|---|---|
| | $R^2$ | RMSE (kg./ha) | $R^2$ | RMSE (kg./ha) | |
| B4 | 0.65 | 38.44 | 0.65 | 38.31 | Yield = −1066.097 (B4) ** + 171.76 |
| B5 | 0.79 | 30.19 | 0.79 | 29.63 | Yield = −1065.86 (B5) * + 209.08 |
| B11 | 0.53 | 46.31 | 0.47 | 45.25 | Yield = −496.74 (B11) * + 204.88 |
| B12 | 0.67 | 37.19 | 0.62 | 40.31 | Yield = −312.63 (B12) ** + 719.05 |

\* $p \leq 0.05$, \*\* $p > 0.05$, B4 = Red, B5 = Red Edge, B11 = SWIR1, B12 = SWIR2.

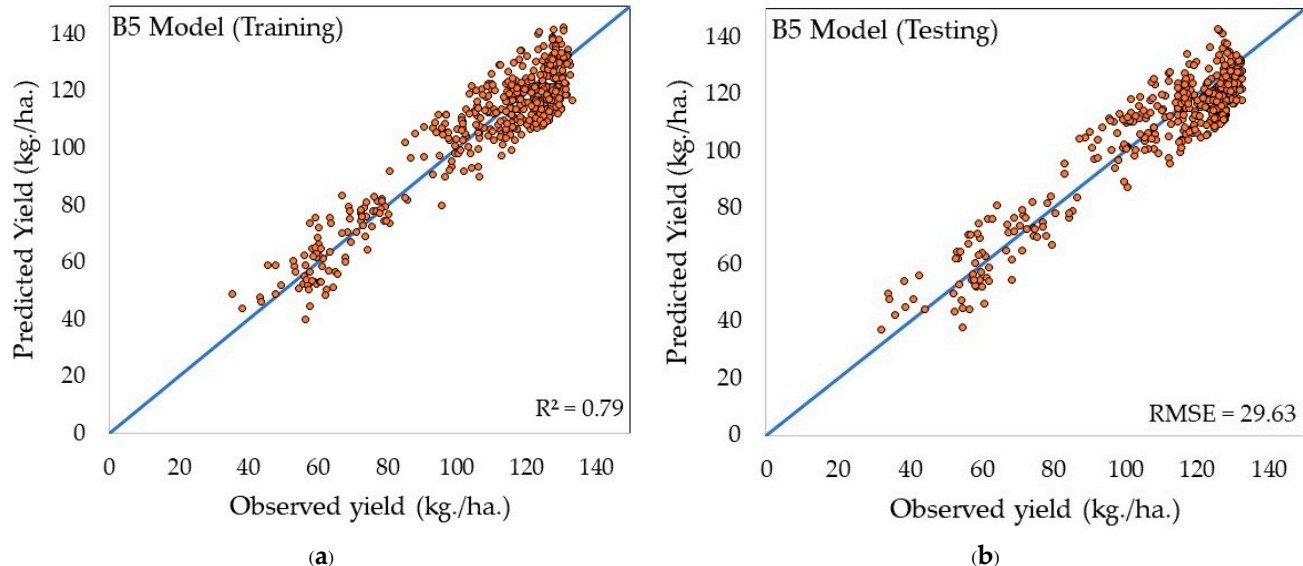

(**a**)                                            (**b**)

**Figure 6.** Scattering plots between observed yield and predicted yield based on the reflectance model (band5-vegetation red edge), which provides the best coefficient of determination. (**a**) Training, (**b**) Testing.

**Table 2.** The $R^2$ and RMSE of linear regression models of the vegetation indices derived from the Sentinel-2 satellite.

| Vegetation Index | Training Model | | Testing Model | | Equation |
|---|---|---|---|---|---|
| | $R^2$ | RMSE (kg./ha) | $R^2$ | RMSE (kg./ha) | |
| NDVI | 0.52 | 44.63 | 0.52 | 44.81 | Yield = 186.39 (NDVI) ** − 11.71 |
| GSAVI | 0.54 | 44.25 | 0.53 | 44.19 | Yield = −743.37 (GSAVI) * + 58.85 |
| NBR | 0.57 | 42.44 | 0.53 | 44.63 | Yield = 171.56 (NBR)) * + 15.74 |
| NR | 0.56 | 43.75 | 0.55 | 43.06 | Yield = −523.58 (NR) * + 181.76 |
| RVI | 0.57 | 43.50 | 0.53 | 43.13 | Yield = −239.89 (RVI) ** + 161.41 |
| MSR | 0.68 | 37.25 | 0.62 | 39.25 | Yield = −1655.81 (MSR) * + 151.17 |

\* $p \leq 0.05$, \*\* $p > 0.05$, Green Soil Adjusted Vegetation Index (GSAVI), Modified Simple Ratio (MSR), Normalized Burn Ratio (NBR), Normalized Difference Vegetation Index (NDVI), Normalized Green (NR), and Ratio Vegetation Index (RVI).

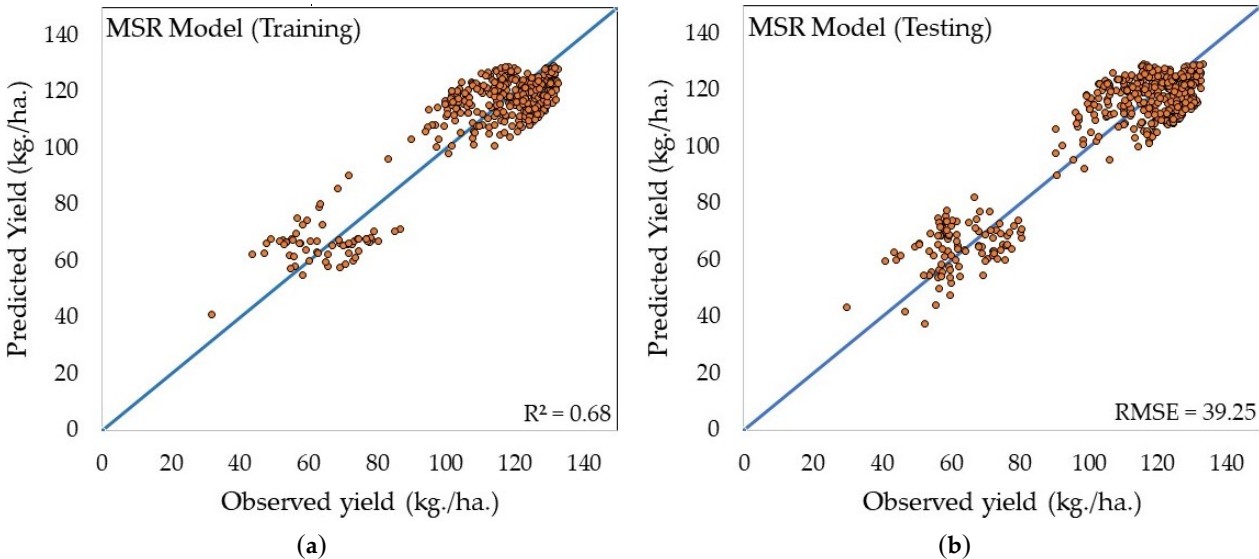

**Figure 7.** Scattering plots Relationship between observed yield and predictive yield. Based on the Vegetation Index (MSR), which provides the best coefficient of determination: (**a**) Data for training; (**b**) Data for testing.

### 3.2.2. Multiple Linear Regression Analysis

The correlation between bands (reflectance) and rubber yield and the correlation between vegetation index and rubber yield results were used as the initial criteria for selecting the bands and vegetation indices as the independent variables for the multiple linear regression process. Trial and error with three categories of independent variables (i.e., spectral band combination, vegetation index, and mixed) was used to build the models. The results showed that the coefficient of determination ($R^2$) and RMSE ranged from 0.70–0.80 and from 29.42–35.75 kg/ha, respectively (Table 3). B5 (vegetation red edge) and MSR provided the best results for the testing data (i.e., $R^2 = 0.80$ and RMSE = 29.42). The scatter plots between observed and predicted yields of band combinations, vegetation index, and mixed are shown in Figures 8–10 respectively. The plots were created by setting the rubber yield from the observed yield as the dependent variable (Y) and the predicted yield as the independent variable (X). Figure 11 shows the map created from the reflectance value and the vegetation index model (B5 and MSR), the best model for 12 months conducted from December 2020 to November 2021.

**Table 3.** Conclusion of the experiment on estimating rubber yield by multiple linear regression Analysis. from the reflectance and vegetation index of the sentinel-2 satellite.

| Model | Training Model | | Testing Model | | Equation |
|---|---|---|---|---|---|
| | $R^2$ | RMSE (kg./ha) | $R^2$ | RMSE (kg./ha) | |
| Band Combinations | 0.79 | 29.94 | 0.79 | 29.69 | Yield = −1124.00 (B5) + 42.33 (B11) + 206.47 |
| Indices | 0.68 | 36.44 | 0.70 | 35.75 | Yield = 70.28 (NBR) − 1184.77 (MSR) + 101.28 |
| B5 and MSR | 0.78 | 29.40 | 0.80 | 29.42 | Yield = −985.79 (B5) − 153.34 (MSR) + 205.51 |

B5 = Red Edge, B11 = SWIR1, Modified Simple Ratio (MSR), Normalized Burn Ratio (NBR).

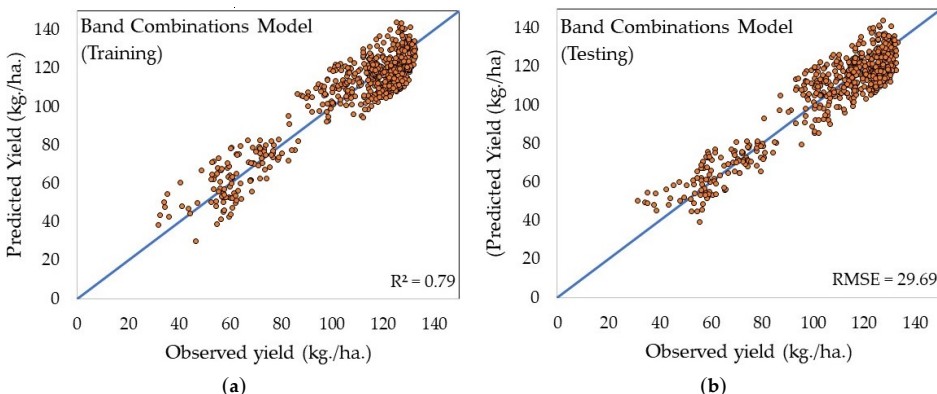

**Figure 8.** Scattering plots Relationship between observed yield and predictive yield. Based on the band combinations model, which provides the best coefficient of determination: (**a**) Data for training; (**b**) Data for testing.

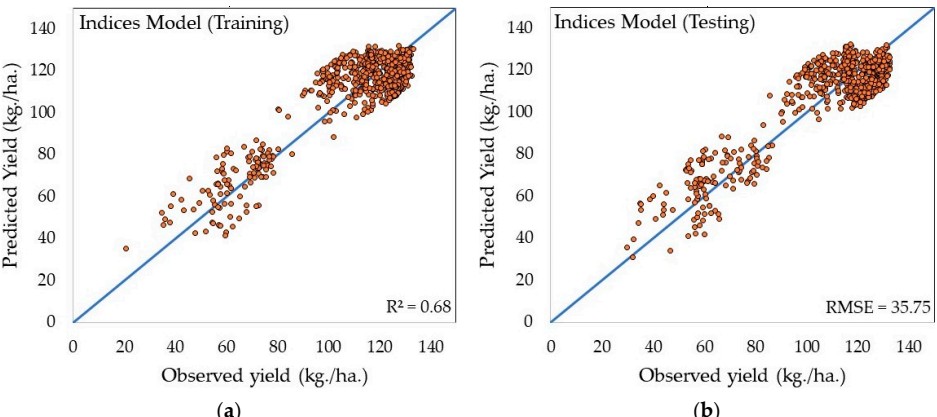

**Figure 9.** Scattering plots relationship between observed yield and predictive yield. Based on the indices model, which provides the best coefficient of determination: (**a**) Data for training; (**b**) Data for testing.

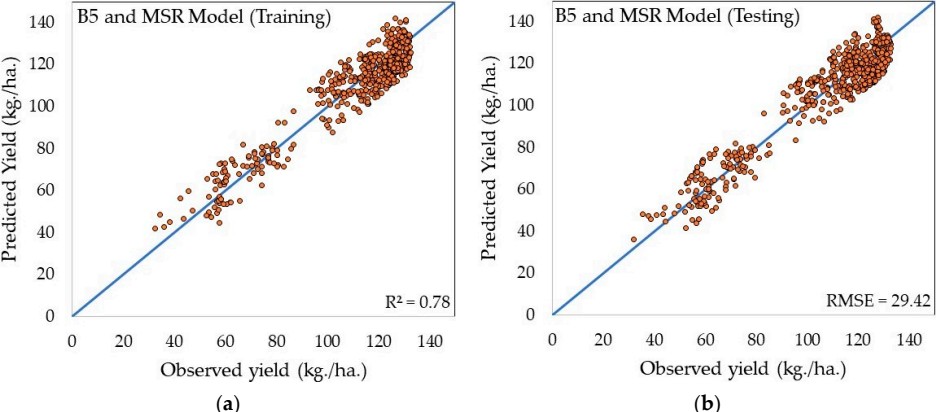

**Figure 10.** Scattering plots Relationship between observed yield and predictive yield. Based on the reflectance value together with the vegetation index model (B5 and MSR), which provides the best coefficient of determination: (**a**) Data for training; (**b**) Data for testing.

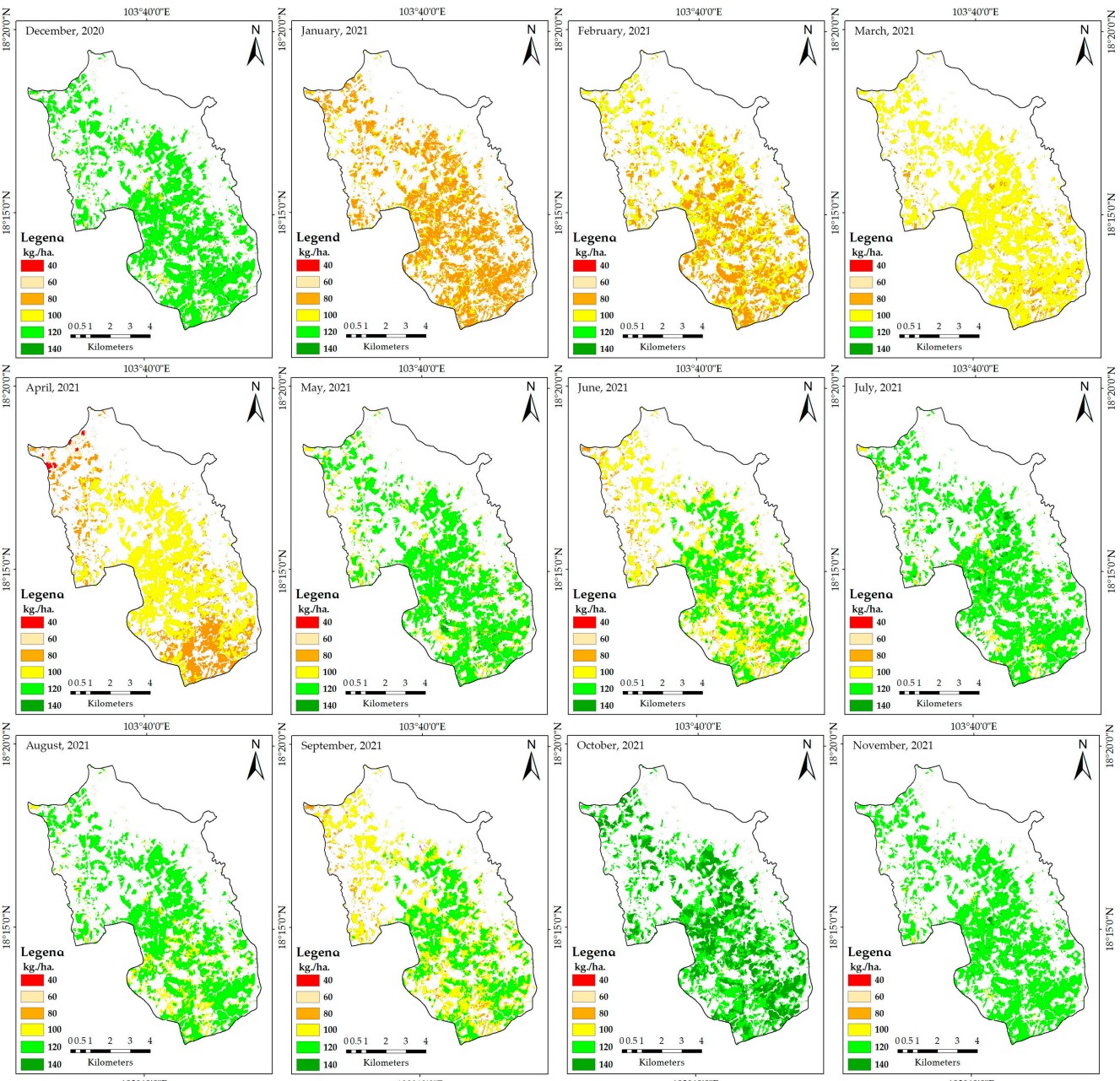

**Figure 11.** Rubber yield model map derived from multiple linear regression analysis from December 2020 to November 2021.

This study showed the potential of using reflectance data and vegetation indices from Sentinel-2 satellite imagery to estimate the yield of rubber plantations in smallholder areas in the Non Somboon Subdistrict, Mueang District, Bueng Kan Province, in northeast Thailand. Similar to previous studies, Sentinel-1 satellites were successfully used to estimate the yield of crops and the oil palm yield model of [45]. This study reveals that the reflectance, i.e., B4 (red), B5 (vegetation red edge), B11 (SWIR1), and B12 ($SWIR^2$) and vegetation index (i.e., MSR) derived from Sentinel-2 satellite imagery are rather well correlated with rubber yield. The vegetation index had a smaller effect on the coefficient of determination ($R^2$) than B5 reflectance (vegetation red edge), especially in dense vegetation conditions like those of the experiments in [23,64]. It seems that B5 (red edge) is the most important reflectance wavelength in estimating rubber yield (Table 1). The B5 (red edge) provided the coefficient of determination ($R^2$), which was higher than the reported value of 0.77 in [60] when they

used the NIR and NDVI in a linear regression model. Previous studies have reported saturation problems in the red and NIR bands [33,65,66].

For vegetation indices, the experimental results show that MSR was the best predictor of the yield of para rubber (Table 2). It was found that the coefficient of determination ($R^2$) was higher than the one reported in [60]. The result is consistent with the study of [67], which reported that MSR had the best correlation with LAI [30] and yield prediction. Although the NDVI shows the greenness of the vegetation, the results show that it is the weakest predictor of rubber yield (Table 2). This is contrary to the report in [60] that NDVI combined with NIR can predict the yield of rubber well. For the mixed model, the results showed that the combination of red-edge reflectance and MSR gave the most accurate yield prediction (lowest RMSE; see Table 3).

The age range of the rubber trees in the study area was 5 to 25 years, and the average was around 12–18 years. The model developed in this study had a high yield error in 5–8-year-old para rubber [68]. Detection is difficult because it is quite difficult to distinguish these particular trees from empty areas or vegetation cover since the trees have fewer leaves and cover only a small part of the total planted area [69]. This problem should be considered in future studies. The red edge-based method demonstrated higher sensitivity in detecting phenological and landscape changes in foliage coloration and vegetation recovery of rubber plantations [68,70]. Although there have been some research studies on the prediction of rubber yield, there have been no studies in smallholder agriculture areas, and most of the previous studies have been done using satellite images with low spatial resolution, such as those from the MODIS satellite (with a 250-m spatial resolution) [57]. This study was conducted with the rubber species RRIM 600. This study was conducted with the rubber species RRIM 600. Field data were collected from 213 smallholder farmers, and rubber plantations are located only in the northeast of Thailand. There may be different environmental factors with others that make for different results. Therefore, in future work, it may be necessary to test in other smallholder areas. Although the results of the prediction of the rubber yield are acceptable and higher than previously reported, there is still an opportunity to improve the accuracy. The use of reflectance and vegetation indices derived from hyperspectral satellite data in combination with meteorological data and the application of complex mathematical models, such as machine learning and deep learning, is interesting for future studies.

## 4. Conclusions

This study aimed to estimate rubber yields in smallholder plantations in the Non Somboon Subdistrict, Mueang District, Bueng Kan Province, in the northeast of Thailand. The reflectance data from Sentinel-2 satellite imagery were acquired for 12 months between December 2020 and November 2021, and 213 plots of data on rubber production in smallholder agriculture were collected. Selected plots were required to have an area of at least 5 rai (0.8 hectares) per plot, and an age range of 5 to 25 years. Farmers' sales invoices providing weight data (number of kilos) were used together with consultation with the relevant authorities in the study area to verify the actual field data. Six vegetation indices, namely Green Soil Adjusted Vegetation Index (GSAVI), Modified Simple Ratio (MSR), Normalized Burn Ratio (NBR), Normalized Difference Vegetation Index (NDVI), Normalized Green (NR), and Ratio Vegetation Index (RVI) were used to estimate the monthly rubber yield. This study revealed that the Sentinel-2 imagery, with its reflectance and vegetation indices, has the potential to be used for rubber tree yield estimation (with a coefficient of determination ($R^2$) of 0.80 and an RMSE of 29.42 kg/ha) in the case of smallholder farms. With respect to spectral reflectance, it was found that the best correlations were B4 (Red), B5 (vegetation red edge), B11 (SWIR), and B12 (SWIR). The vegetation indices showed that the best correlations with rubber yield were the Normalized Burn Ratio (NBR), Normalized Green (NR), and Modified Simple Ratio (MSR). The red edge wavelength (band 5) yielded the best prediction with $R^2$ = 0.79 and RMSE = 29.63 kg/ha, outperforming all other spectral bands and VIs. Regarding vegetation indices, the MSR index delivered

the highest coefficient of determination, with $R^2$ = 0.62 and RMSE = 39.25 kg/ha. When the red edge reflectance was combined with the MSR, the prediction model only slightly improved, with a coefficient of determination ($R^2$) of 0.80 and an RMSE of 29.42 kg/ha.

**Author Contributions:** Conceptualization, N.B., J.N., N.S., W.K. and S.K.; methodology, N.B., J.N., N.S., W.K. and S.K.; validation, N.B. and S.K.; formal analysis, N.B. and S.K.; investigation, N.B. and S.K.; writing—original draft preparation, N.B., W.K. and S.K.; writing—review and editing, N.B., W.K. and S.K.; supervision, N.B., W.K., A.H. and S.K. All authors have read and agreed to the published version of the manuscript.

**Funding:** This research project was financially supported by Mahasarakham University. (Grant Agreement Number: 6617002/2566).

**Institutional Review Board Statement:** Not applicable.

**Informed Consent Statement:** Not applicable.

**Data Availability Statement:** This study did not report any data.

**Acknowledgments:** The author would like to thank the rubber plantation agriculture. Non Somboon Subdistrict, Mueang District, Bueng Kan Province, for supporting data in this study.

**Conflicts of Interest:** The authors declare no conflict of interest.

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
