# Peer review of "Estimation of Rubber Yield Using Sentinel-2 Satellite Data"

_sustainability, doi:10.3390/su15097223_

Round 1

Reviewer 1 Report

This paper presents a study aimed to estimate rubber yields in smallholder plantations, that was monitored with Sentinel-2 satellite imagery, in a very interesting research project. Estimation of rubber yield was done by multiple linear regression analysis and results are important.

Author Response

Manuscript update 20230331

Reviewer 2 Report

1- The conclusion section is unclear and does not fully correspond to the study's goal. Rewrite the conclusion so that it is clearly related to the study's objectives.

2- In their research, the authors used a Sentinel-2 satellite image. Why? What makes this satellite sensor different from others in terms of image quality or resolution?

3- The image chosen for this study has a resolution of 20 m. Do you consider that a low resolution? Wasn’t it possible to improve that resolution to a higher one to improve the results?

4- The introduction section was overly lengthy and filled with extraneous details, primarily concerning rubber and the surrounding area. Reduced the section down to the essentials of your research.

5- Could the authors provide an explanation for why the image was taken between 2019 and 2020? Why not update your study to say, 2021–2022, for example?

6- Which Sentinel-2 channel captured the image? Provide for your study.

7- Figures 6–9 in black and white. For the best visualization and details, I advised using color mode.

Author Response

Dear Reviewer

Reviewer 2:

  1. 1. The conclusion section is unclear and does not fully correspond to the study's goal. Rewrite the conclusion so that it is clearly related to the study's objectives.

Thanks for your kind suggestions, which are valuable for improving the manuscript. We have rewritten the conclusion part, in the new manuscript (line 303-326) the following: 

This study aimed to estimate rubber yields in smallholder plantations the area in Non Somboon Subdistrict, Mueang District, Bueng Kan Province, in the northeast of Thailand. The reflectance data from Sentinel-2 satellite imagery were acquired for 12 months between December 2020 and November 2021, and 213 plots of data on rubber production in smallholder agriculture were collected. Selected plots were required to have an area of at least 5 rai (0.8 hectares) per plot, and an age range of 5 to 25 years. Farmers’ sales invoices providing weight data (number of kilos) were used together with consultation with the relevant authorities in the study area to verify the actual field data. Six vegetation indices, namely Green Soil Adjusted Vegetation Index (GSAVI), Modified Simple Ratio (MSR), Normalized Burn Ratio (NBR), Normalized Difference Vegetation Index (NDVI), Normalized Green (NR) and Ratio Vegetation Index (RVI) were used to estimate the monthly rubber yield. This study revealed that the Sentinel-2 imagery, with its reflectance and vegetation indices, has the potential to be used for rubber tree yield estimation (with a coefficient of determination (R2) of 0.80 and an RMSE of 29.42 kg/ha). With respect to spectral reflectance, it was found that the best correlations were B4 (Red), B5 (vegetation red edge), B11 (SWIR), and B12 (SWIR). The vegetation indices showed that the best correlations with rubber yield were Normalized Burn Ratio (NBR), Normalized Green (NR), and Modified Simple Ratio (MSR). The red edge wavelength (band 5) yielded the best prediction with R2 = 0.79 and RMSE = 29.63 kg/ha, outperforming all other spectral bands and VIs. In terms of vegetation indices, the MSR index delivered the highest coefficient of determination, with R2 = 0.62 and RMSE = 39.25 kg/ha. When the red edge reflectance was combined with the MSR, the prediction model only slightly improved, with a coefficient of determination (R2) of 0.80 and an RMSE of 29.42 kg/ha.

  1. 2. In their research, the authors used a Sentinel-2 satellite image. Why? What makes this satellite sensor different from others in terms of image quality or resolution?

Thank you very much for your valuable question regarding the use of the Sentinel-2 satellite image we proposed for use in this study. According to the previous literature review, it has been found that the use of satellite imagery to estimate rubber yield has not been very effective, and the spatial resolution is low.  The satellite is equipped with an optoelectronic multispectral sensor for surveying with a sentinel-2 resolution of 10 to 60 m. in the visible, near infrared (VNIR), and shortwave infrared (SWIR) spectral zones, including 13 spectral channels, which ensures the capture of differences in vegetation state, including temporal changes, and also minimizes impact on the quality of atmospheric photography. In addition, the Sentinel-2 satellite will send images back to the ground every five days, providing imagery faster than other satellites.

We also revised the manuscript to explain the reasons in the revised manuscript from line 90-102 as the following:

However, most studies utilized low-resolution (i.e., MODIS), and a study used moder-ate-resolution but low temporal resolution (i.e., THEOS) satellites for rubber yield ap-plications, which is not reasonable for smallholder farms with small plot sizes (less than 4 hectares of land). It is found that the satellite data is usually used for applications in the rubber industry; however, the use of satellite imagery to estimate rubber yield has not been very effective, and the satellites used have low spatial resolution [59]. The Sentinel-2 satellites have a high resolution that allows them to distinguish between rubber plantations and complex natural forests [57]. Since the Sentinel-2A and Senti-nel-2B satellites work together, they have high temporal resolution and optimum spatial resolution, allowing the utilization of time series data [58]. May be suitable for tasks that require continuous monitoring and small areas. Various research has focused on increasing the production of natural rubber in a wide area but has not considered the constraints of smallholder primary producers.

  1. 3. The image chosen for this study has a resolution of 20 m. Do you consider that a low resolution? Wasn’t it possible to improve that resolution to a higher one to improve the results?

Thank you very much for your question regarding the use of a resolution of 20 m. The Sentinel-2 satellite image we proposed for use in this study was chosen because the Sentinel-2 satellite images are presented as free data, are convenient to use for study, and can actually be used in Thailand, so we avoided using the costly resolution image data. However, we have discussed the use of reflectance and vegetation index obtained from hyperspectral satellite data as well as the application of complex mathematical models for future studies.

  1. 4. The introduction section was overly lengthy and filled with extraneous details, primarily concerning rubber and the surrounding area. Reduced the section down to the essentials of your research.

Thanks for your kind suggestions, which are valuable for improving the manuscript. we have rewritten the introduction. (Line 35-119) the following:

Rubber (Hevea brasiliensis) is a perennial plant grown for the production of natural rubber. It also helps to absorb carbon dioxide stored in the form of biomass [1]. There are over 11 million hectares of agricultural land worldwide, with rubber plantations covering about 9.2 million hectares (78%) in Southeast Asia, about 3.67 million hectares (31%) in Indonesia, and about 3.23 million hectares in Thailand [2]. Most rubber trees are grown in the tropics and have an economic life of 30 to 35 years [3]. The rubber trees are cut down to provide timber for trade; every year, about 3-4% of the rubber plantation area is cleared to be replaced by a new generation of rubber [4, 5]. Last century, rubber trees were planted over large areas, currently, rubber cultivation is gradually shifting to the smallholder sector [6]. Most Southeast Asian countries converted to smallholder agriculture in 2018, and contribute more than three-quarters of the world's natural rubber production, with the main natural rubber producing countries being Thailand, Indonesia, Vietnam, India, China and Malaysia. In 2018, natural rubber production reached 14.33 million tons [2]. Thailand, of course, is located in the tropics, where the environment is suitable for the cultivation of rubber. In the past, the cultivation of rubber in Thailand was popular in the southern and eastern regions; however, the rate of expansion of rubber plantations in the southern region has begun to decline. With policies aimed at reducing the amount of land unsuitable for planting rubber trees and allowing farmers to plant oil palms or fruit trees instead, rubber plantations have recently expanded to the northeast, where they had never been planted before [7]. It is obvious that rubber is a significant commercial crop for Thailand.

Natural rubber production is vital to more than 20 million farmers worldwide. Over the last century, rubber cultivation has gradually shifted to the smallholder [8], and this sector accounts for more than 75% of the world's natural rubber production [9, 10]. The size of rubber plantations is assessed differently by each country; however, it is reported that most smallholder farmers have less than 4 hectares of land [11, 12]. Smallholders in Malaysia, Thailand, Myanmar, India, and Indonesia produced natural rubber at a high rate (i.e., between 85% and 93%) [13]. However, smallholder rubber farmers continue to have limited income and limited access to financial resources [14, 15]. Therefore, creating sustainability in natural rubber production, especially for smallholders, is crucial for the development of the global economy.

Remote sensing technology can effectively provide large spatial data [16-20]. Satellite imagery sources are often freely available, they cover a large geographic area, and have high temporal resolution. Since satellite images have multiple spectra, different photo indices are presented to distinguish harvesting areas [21], and they play an important role in mapping local and regional rubber plantations [22-24]. Furthermore, these tools have been used to provide real-time understanding of changes in rubber plantation area [25, 26]. This will reduce the time, labor and cost of inspecting rubber plantations in the field, where access is limited [27, 28]. Because estimating yields is important to maintaining price stabilization, many countries use common techniques to collect yield data and area-based field reports. Most of these data come from post-harvest surveys that are conducted relatively late [29]. Satellite remote sensing is used for a variety of problems and applications [30], including yield estimation and agricultural yield prediction. It is an interesting research work and of great importance [28]. Agricultural yield assessment and forecasting comprise an essential component [1, 31]. Several studies have utilized vegetation indices to estimate the yield of crops such as rice, wheat, barley [32-36], potato [37-39], maize [40-44], and oil palm [45-49]. Obviously, the application of remote sensing in agriculture, including yield estimation and crop forecasting, is of great research interest and importance [34, 38, 50, 51]. Several studies utilized low-resolution MODIS satellites to solve the problem of classification based on biophysical properties from time series, which provides good performance [23, 34, 52]. The Landsat satellites have moderate spatial resolution (i.e., 30-meter spatial resolution) and have been accurately used for estimating rubber tree growth and age [53-56]. The THEOS satellite, in particular, has been used to analyze the change and expansion of the rubber tree area and the rate of change in northeast Thailand [7]. However, most studies utilized low-resolution (i.e., MODIS), and a study used moder-ate-resolution but low temporal resolution (i.e., THEOS) satellites for rubber yield ap-plications, which is not reasonable for smallholder farms with small plot sizes (less than 4 hectares of land). It is found that the satellite data is usually used for applications in the rubber industry; however, the use of satellite imagery to estimate rubber yield has not been very effective, and the satellites used have low spatial resolution [59]. The Sentinel-2 satellites have a high resolution that allows them to distinguish between rubber plantations and complex natural forests [57]. Since the Sentinel-2A and Senti-nel-2B satellites work together, they have high temporal resolution and optimum spatial resolution, allowing the utilization of time series data [58]. May be suitable for tasks that require continuous monitoring and small areas. Various research has focused on increasing the production of natural rubber in a wide area, but has not considered the constraints of smallholder primary producers. In addition, it has not been clearly stated how the information on rubber yields for smallholders will be collected [60]. Therefore, this study aims to test the Sentinel-2 satellite in the application of rubber yield estimation for smallholder farms. Both spectral and vegetation indices from Sentinel-2 were used as inputs to the models. Linear and multiple linear regressions were used to build prediction models. Each model was statistically compared to determine the most suitable model for rubber monthly yield prediction. The accurate model will be helpful to stabilize the country's rubber prices in the future.

  1. 5. Could the authors provide an explanation for why the image was taken between 2019 and 2020? Why not update your study to say, 2021–2022, for example?

Thank you very much for your question regarding of why the use of image was taken between 2019 and 2020. that we present for use in this study. In this study, there was a limitation in funding for the study area. The study started in 2020 and was scheduled to go to the sampling area to conduct the experiment for 12 months.

  1. 6. Which Sentinel-2 channel captured the image? Provide for your study.

Thank you very much for your question about the Sentinel-2 channel captured in this study. This study used 13 bands of Sentinel-2 and Vis derived from it. We have provided the channels use in this study at Line 181-182, following:

“The correlations between bands (reflectance) Correlation coefficient (R). For the spectral reflection of 13 bands, it was found that the best correlations were B4 (Red), B5 (vegetation red edge), B11 (SWIR), and B12 (SWIR). “

  1. 7. Figures 6–9 in black and white. For the best visualization and details, I advised using color mode.

Thanks for the reviewer’s comment. We have revised the map in figure 6-9 already.

(a)

(b)

Figure 6. Scattering plots between observed yield and predicted yield based on the reflectance model (band5-vegetation red edge), which provides the best coefficient of determination. (a) training, (b) testing.

(a)

(b)

Figure 7. Scattering plots Relationship between observed yield and predictive yield. Based on the Vegetation Index (MSR), which provides the best coefficient of determination: (a) data for training; (b) data for testing.

(a)

(b)

Figure 8. Scattering plots Relationship between observed yield and predictive yield. Based on the band combinations model, which provides the best coefficient of determination: (a) data for training; (b) data for testing.

(a)

(b)

Figure 9. Scattering plots relationship between observed yield and predictive yield. Based on the indices model, which provides the best coefficient of determination: (a) data for training; (b) data for testing.

 v

(a)

(b)

Figure 10. Scattering plots Relationship between observed yield and predictive yield. Based on the reflectance value together with the vegetation index model (B5 and MSR), which provides the best coefficient of determination: (a) data for training; (b) data for testing.

Reviewer 3 Report

1. Include the conclusion in the abstract

2. Keywords should not match the title. revise the keywords

3. Line number 49- suddenly you are mentioning Brazil. Please look into the sentence and rewrite accordingly.

4. line 57- Arrange the reference as per journal format

5. Rewrite the Introduction part, at present, it is not giving any meaning to your study

6. Highlight the Novelty and originality of the study

7. Material and methods section are written well. 

8. Sample size (213 farms) looks less, justify with your findings

9. The R2 value in both the training model and testing model is low. Justify

10. Include limitations and implications of the study in the discussion section 

11. Rewrite the conclusion section, as I am not finding difference in abstract and conclusion

Author Response

Reviewer 3:

  1. 1. Include the conclusion in the abstract.

               Thanks for your kind suggestions, which are valuable for improving the manuscript. We have rewritten the abstract part, in the new manuscript (line 22-31) the following:

The study found that the red edge spectral band (band 5) provided the best prediction with R2 = 0.79 and RMSE = 29.63 kg/ha, outperforming all other spectral bands and VIs. The MSR index provided the highest coefficient of determination, with R2 = 0.62 and RMSE = 39.25 kg/ha. When the red edge reflectance was combined with the best VI, MSR, the prediction model only slightly improved, with a coefficient of determination (R2) of 0.80 and an RMSE of 29.42 kg/ha. The results demonstrated that the Sentinel-2 data are suitable for rubber yield prediction for smallholder farmers. The findings of this study can be used as a guideline to apply in other countries or areas. Future studies will require the use of reflectance and vegetation indices derived from satellite data in combination with meteorological data, as well as the application of complex models such as machine learning and deep learning.

  1. 2. Keywords should not match the title. revise the keywords.

Thanks for your kind suggestions, which are valuable for improving the manuscript. We have rewritten the keywords part, in the new manuscript (line 32) the following: natural rubber; smallholder; Sentinel-2; yield estimation model; reflectance.

  1. 3. Line number 49- suddenly you are mentioning Brazil. Please look into the sentence and rewrite accordingly.

Thank you very much for the reviewer’s comment, which are valuable for improving the manuscript. We have removed this section due to recommendations from other reviewers.

  1. 4. line 57- Arrange the reference as per journal format.

                    Thank you very much for the reviewer’s comment, which are valuable for improving the manuscript. We have removed this section due to recommendations from other reviewers and have checked all mistake.

  1. 5. Rewrite the Introduction part, at present, it is not giving any meaning to your study.

                    Thanks for your kind suggestions, which are valuable for improving the manuscript. we have rewritten the introduction. (Line 35-119) the following:

               Rubber (Hevea brasiliensis) is a perennial plant grown for the production of natural rubber. It also helps to absorb carbon dioxide stored in the form of biomass [1]. There are over 11 million hectares of agricultural land worldwide, with rubber plantations covering about 9.2 million hectares (78%) in Southeast Asia, about 3.67 million hectares (31%) in Indonesia, and about 3.23 million hectares in Thailand [2]. Most rubber trees are grown in the tropics and have an economic life of 30 to 35 years [3]. The rubber trees are cut down to provide timber for trade; every year, about 3-4% of the rubber plantation area is cleared to be replaced by a new generation of rubber [4, 5]. Last century, rubber trees were planted over large areas, currently, rubber cultivation is gradually shifting to the smallholder sector [6]. Most Southeast Asian countries converted to smallholder agriculture in 2018, and contribute more than three-quarters of the world's natural rubber production, with the main natural rubber producing countries being Thailand, Indonesia, Vietnam, India, China and Malaysia. In 2018, natural rubber production reached 14.33 million tons [2]. Thailand, of course, is located in the tropics, where the environment is suitable for the cultivation of rubber. In the past, the cultivation of rubber in Thailand was popular in the southern and eastern regions; however, the rate of expansion of rubber plantations in the southern region has begun to decline. With policies aimed at reducing the amount of land unsuitable for planting rubber trees and allowing farmers to plant oil palms or fruit trees instead, rubber plantations have recently expanded to the northeast, where they had never been planted before [7]. It is obvious that rubber is a significant commercial crop for Thailand.

Natural rubber production is vital to more than 20 million farmers worldwide. Over the last century, rubber cultivation has gradually shifted to the smallholder [8], and this sector accounts for more than 75% of the world's natural rubber production [9, 10]. The size of rubber plantations is assessed differently by each country; however, it is reported that most smallholder farmers have less than 4 hectares of land [11, 12]. Smallholders in Malaysia, Thailand, Myanmar, India, and Indonesia produced natural rubber at a high rate (i.e., between 85% and 93%) [13]. However, smallholder rubber farmers continue to have limited income and limited access to financial resources [14, 15]. Therefore, creating sustainability in natural rubber production, especially for smallholders, is crucial for the development of the global economy.

Remote sensing technology can effectively provide large spatial data [16-20]. Satellite imagery sources are often freely available, they cover a large geographic area, and have high temporal resolution. Since satellite images have multiple spectra, different photo indices are presented to distinguish harvesting areas [21], and they play an important role in mapping local and regional rubber plantations [22-24]. Furthermore, these tools have been used to provide real-time understanding of changes in rubber plantation area [25, 26]. This will reduce the time, labor and cost of inspecting rubber plantations in the field, where access is limited[27, 28]. Because estimating yields is important to maintaining price stabilization, many countries use common techniques to collect yield data and area-based field reports. Most of these data come from post-harvest surveys that are conducted relatively late [29]. Satellite remote sensing is used for a variety of problems and applications [30], including yield estimation and agricultural yield prediction. It is an interesting research work and of great importance [28]. Agricultural yield assessment and forecasting comprise an essential component [1, 31]. Several studies have utilized vegetation indices to estimate the yield of crops such as rice, wheat, barley [32-36], potato [37-39], maize [40-44], and oil palm [45-49]. Obviously, the application of remote sensing in agriculture, including yield estimation and crop forecasting, is of great research interest and importance [34, 38, 50, 51]. Several studies utilized low-resolution MODIS satellites to solve the problem of classification based on biophysical properties from time series, which provides good performance [23, 34, 52]. The Landsat satellites have moderate spatial resolution (i.e., 30 meter spatial resolution) and have been accurately used for estimating rubber tree growth and age [53-56]. The THEOS satellite, in particular, has been used to analyze the change and expansion of the rubber tree area and the rate of change in northeast Thailand [7]. However, most studies utilized low-resolution (i.e., MODIS), and a study used moderate-resolution but low temporal resolution (i.e., THEOS) satellites for rubber yield applications, which is not reasonable for smallholder farms with small plot sizes (less than 4 hectares of land). It is found that the satellite data is usually used for applications in the rubber industry; however, the use of satellite imagery to estimate rubber yield has not been very effective, and the satellites used have low spatial resolution [59]. The Sentinel-2 satellites have a high resolution that allows them to distinguish between rubber plantations and complex natural forests [57]. Since the Sentinel-2A and Sentinel-2B satellites work together, they have high temporal resolution and optimum spatial resolution, allowing the utilization of time series data [58]. May be suitable for tasks that require continuous monitoring and small areas. Various research has focused on increasing the production of natural rubber in a wide area, but has not considered the constraints of smallholder primary producers. In addition, it has not been clearly stated how the information on rubber yields for smallholders will be collected [60]. Therefore, this study aims to test the Sentinel-2 satellite in the application of rubber yield estimation for smallholder farms. Both spectral and vegetation indices from Sentinel-2 were used as inputs to the models. Linear and multiple linear regressions were used to build prediction models. Each model was statistically compared to determine the most suitable model for rubber monthly yield prediction. The accurate model will be helpful to stabilize the country's rubber prices in the future.

  1. 6. Highlight the Novelty and originality of the study.

               Thank you very much for the reviewer’s comment, which are valuable for improving the manuscript. We have rewritten in the new manuscript (line 90-109) the following:

However, most studies utilized low-resolution (i.e., MODIS), and a study used moder-ate-resolution but low temporal resolution (i.e., THEOS) satellites for rubber yield ap-plications, which is not reasonable for smallholder farms with small plot sizes (less than 4 hectares of land). It is found that the satellite data is usually used for applications in the rubber industry; however, the use of satellite imagery to estimate rubber yield has not been very effective, and the satellites used have low spatial resolution [59]. The Sentinel-2 satellites have a high resolution that allows them to distinguish between rubber plantations and complex natural forests [57]. Since the Sentinel-2A and Senti-nel-2B satellites work together, they have high temporal resolution and optimum spatial resolution, allowing the utilization of time series data [58]. May be suitable for tasks that require continuous monitoring and small areas. Various research has focused on increasing the production of natural rubber in a wide area but has not considered the constraints of smallholder primary producers. In addition, it has not been clearly stated how the information on rubber yields for smallholders will be collected [60]. Therefore, this study aims to test the Sentinel-2 satellite in the application of rubber yield estimation for smallholder farms. Both spectral and vegetation indices from Sentinel-2 were used as inputs to the models. Linear and multiple linear regressions were used to build prediction models. Each model was statistically compared to determine the most suitable model for rubber monthly yield prediction. The accurate model will be helpful to stabilize the country's rubber prices in the future.    

And revised the conclusion:

This study revealed that the Sentinel-2 imagery, with its reflectance and vegetation in-dices, has the potential to be used for rubber tree yield estimation (with a coefficient of determination (R2) of 0.80 and an RMSE of 29.42 kg/ha) in case of smallholder farm. With respect to spectral reflectance, it was found that the best correlations were B4 (Red), B5 (vegetation red edge), B11 (SWIR), and B12 (SWIR). The vegetation indices showed that the best correlations with rubber yield were Normalized Burn Ratio (NBR), Normalized Green (NR), and Modified Simple Ratio (MSR). The red edge wavelength (band 5) yielded the best prediction with R2 = 0.79 and RMSE = 29.63 kg/ha, outperforming all other spectral bands and VIs. In terms of vegetation indices, the MSR index delivered the highest coefficient of determination, with R2 = 0.62 and RMSE = 39.25 kg/ha. When the red edge reflectance was combined with the MSR, the prediction model only slightly improved, with a coefficient of determination (R2) of 0.80 and an RMSE of 29.42 kg/ha.

  1. 7. Material and methods section are written well.

               Thank you very much for the reviewer’s comment.

  1. 8. Sample size (213 farms) looks less, justify with your findings.

               Thank you very much for your question regarding of why the use of Sample size (213 farms). In this study, our proposed sample size (213 farms) is sufficient because the sample size per plot is very large and the age of the rubber is about the same size have similar temperatures. The total study area is 3,025 hectares which has more study area than the work of [64] with an area of only 148 hectares.

  1. 9. The R2 value in both the training model and testing model is low. Justify

               Thank you very much for the reviewer’s comment, which are valuable for improving the manuscript. However, this study evaluated rubber yield over a large area with the Sentinel-2 satellite having better resolution than the MODIS satellite [67]. It also gave a better R2 value than [64]'s work. with R2 = 0.62 for linear regression analysis and R2 = 0.76 for multiple linear regression analysis. But this study gave a higher value of R2 than either type. We have revised in the discussion part Line 261-273.

  1. 10. Include limitations and implications of the study in the discussion section.

Thanks for your kind suggestions, which are valuable for improving the manuscript. we have rewritten the introduction. (Line 288-2971) the following:

This study was conducted with the rubber species RRIM 600. This study was conducted with the rubber species RRIM 600. Field data were collected from 213 smallholder farmers, and rubber plantations are located only in the northeast of Thailand. There may be different environmental factors with others that make for different results. Therefore, in future work, it may be necessary to test in other smallholder areas. Although the results of the prediction of the rubber yield are acceptable and higher than previously reported, there is still an opportunity to improve the accuracy. The use of reflectance and vegetation indices derived from hyperspectral satellite data in combination with meteorological data and the application of complex mathematical models such as ma-chine learning and deep learning is interesting for future studies.

  1. 11. Rewrite the conclusion section, as I am not finding difference in abstract and conclusion.

               Thanks for your kind suggestions, which are valuable for improving the manuscript. we have rewritten the conclusion. (Line 299-321):

This study aimed to estimate rubber yields in smallholder plantations the area in Non Somboon Subdistrict, Mueang District, Bueng Kan Province, in the northeast of Thailand. The reflectance data from Sentinel-2 satellite imagery were acquired for 12 months between December 2020 and November 2021, and 213 plots of data on rubber production in smallholder agriculture were collected. Selected plots were required to have an area of at least 5 rai (0.8 hectares) per plot, and an age range of 5 to 25 years. Farmers’ sales invoices providing weight data (number of kilos) were used together with consultation with the relevant authorities in the study area to verify the actual field data. Six vegetation indices, namely Green Soil Adjusted Vegetation Index (GSAVI), Modified Simple Ratio (MSR), Normalized Burn Ratio (NBR), Normalized Difference Vegetation Index (NDVI), Normalized Green (NR) and Ratio Vegetation Index (RVI) were used to estimate the monthly rubber yield. This study revealed that the Sentinel-2 imagery, with its reflectance and vegetation indices, has the potential to be used for rubber tree yield estimation (with a coefficient of determination (R2) of 0.80 and an RMSE of 29.42 kg/ha) in case of smallholder farm. With respect to spectral reflectance, it was found that the best correlations were B4 (Red), B5 (vegetation red edge), B11 (SWIR), and B12 (SWIR). The vegetation indices showed that the best correlations with rubber yield were Normalized Burn Ratio (NBR), Normalized Green (NR), and Modified Simple Ratio (MSR). The red edge wavelength (band 5) yielded the best prediction with R2 = 0.79 and RMSE = 29.63 kg/ha, outperforming all other spectral bands and VIs. In terms of vegetation indices, the MSR index delivered the highest co-efficient of determination, with R2 = 0.62 and RMSE = 39.25 kg/ha. When the red edge reflectance was combined with the MSR, the prediction model only slightly improved, with a coefficient of determination (R2) of 0.80 and an RMSE of 29.42 kg/ha.

Reviewer 4 Report

Thanks to given me an opportunity to review the article “ Estimation of Rubber Yield using Sentinel-2 Satellite Data”. The manuscript is good and well written have some positive information but required improvement.

Introduction: It needs to develop some hypothesis as well as research questions just stating the aim is not a novel idea to represent your study. Authors must follow the following article and develop the objectives like this  i.e,  In this study we employed age class mapping from I extremely high resolution data accessible on Google Earth as well as additional indications from three sources. Here, the presence of pits indicates the change from natural land cover to a rubber plantation. Moreover, a 2019 GEDI tree height map is used to distinguish between field crops and tree cover (Potapov et al., 2021). In order to comprehend the phenological fluctuation of the various plants, MODIS EVI (10-day) data were employed.

Methods:

Authors must include Sentinel-2A and Thematic Mapper (TM), ETM, ETM +, and Thematic Mapper (TM) datasets from the Landsat series were utilised in this study to locate rubber plantations and examine the dynamics of the rubber plantation. Phenology‑based rubber mapping (dNDVI) and

MODIS‑EVI, Vegetation tree height (lidar) based discrimination

Results:

Authors must have to map the spatial distribution of rubber plantation, Characterizing the rubber stand age.

Discussion

Change the discussion according to findings.

Author Response

Reviewer 4: Thanks to given me an opportunity to review the article “Estimation of Rubber Yield using Sentinel-2 Satellite Data”. The manuscript is good and well written have some positive information but required improvement.

  1. 1. Introduction: It needs to develop some hypothesis as well as research questions just stating the aim is not a novel idea to represent your study. Authors must follow the following article and develop the objectives like this i.e, In this study we employed age class mapping from I extremely high resolution data accessible on Google Earth as well as additional indications from three sources. Here, the presence of pits indicates the change from natural land cover to a rubber plantation. Moreover, a 2019 GEDI tree height map is used to distinguish between field crops and tree cover (Potapov et al., 2021). In order to comprehend the phenological fluctuation of the various plants, MODIS EVI (10-day) data were employed.

Thanks for your kind suggestions, which are valuable for improving the manuscript. we have rewritten the Introduction. (Line 93-109) the following:           

It is found that the satellite data is usually used for applications in the rubber industry; however, the use of satellite imagery to estimate rubber yield has not been very effective, and the satellites used have low spatial resolution [59]. The Sentinel-2 satellites have a high resolution that allows them to distinguish between rubber plantations and complex natural forests [57]. Since the Sentinel-2A and Sentinel-2B satellites work together, they have high temporal resolution and optimum spatial resolution, allowing the utilization of time series data [58]. May be suitable for tasks that require continuous monitoring and small areas. Various research has focused on increasing the production of natural rubber in a wide area but has not considered the constraints of smallholder primary producers. In addition, it has not been clearly stated how the information on rubber yields for smallholders will be collected [60]. Therefore, this study aims to test the Sentinel-2 satellite in the application of rubber yield estimation for smallholder farms. Both spectral and vegetation indices from Sentinel-2 were used as inputs to the models. Linear and multiple linear regressions were used to build prediction models. Each model was statistically compared to determine the most suitable model for rubber monthly yield prediction. The accurate model will be helpful to stabilize the country's rubber prices in the future.

  1. 2. Methods: Authors must include Sentinel-2A and Thematic Mapper (TM), ETM, ETM +, and Thematic Mapper (TM) datasets from the Landsat series were utilised in this study to locate rubber plantations and examine the dynamics of the rubber plantation. Phenology‑based rubber mapping (dNDVI) and MODIS‑EVI, Vegetation tree height (lidar) based discrimination.

Thank you for the valuable suggestion. Due to this study aim to utilized the sentinel-2 data for rubber yield estimation in smallholder farm in Thailand. So, other images seem to be out of our scope. However, it is very interesting for the future study to compare the several satellite data.

-

  1. 3. Results: Authors must have to map the spatial distribution of rubber plantation, Characterizing the rubber stand age.

Thanks for the reviewer’s comment. We have already added the map in Figure 11.

Figure 11. Rubber yield model map derived from multiple linear regression analysis from December 2020 to November 2021.

  1. 4. Discussion: Change the discussion according to findings.

               Thank you very much for the reviewer’s comment, which are valuable for improving the manuscript. We have rewritten in the new manuscript (line 252-297) the following:

This study showed the potential of use reflectance data and vegetation indices from Sentinel-2 satellite imagery to estimate the yield of rubber plantations in smallholder areas in Non Somboon Subdistrict, Mueang District, Bueng Kan Province, in the northeast of Thailand. Similar to previous studies, Sentinel-1 satellites were successfully used to estimate the yield of crops and the oil palm yield model of [45]. This study reveals that the reflectance (i.e., B4 (red), B5 (vegetation red edge), B11 (SWIR1), and B12 (SWIR2)) and vegetation index (i.e., MSR) derived from Sentinel-2 satellite imagery are rather well correlated with rubber yield. The vegetation index had a smaller effect on the coefficient of determination (R2) than B5 reflectance (vegetation red edge), especially in dense vegetation conditions like those of the experiments in [23, 64].  It seems that B5 (red edge) is the most important reflectance wavelength in the estimation of rubber yield (Table 1). The B5 (red edge) provided the coefficient of determination (R2), which was higher than the reported value of 0.77 in [60] when they used the NIR and NDVI in a linear regression model. Previous studies have reported saturation problems in the red and NIR bands [33, 65, 66].

For vegetation indices, the experimental results show that MSR was the best predictor of the yield of para rubber (Table 2), It was found that the coefficient of determination (R2) was higher than the reported in [60]. The result is consistent with the study of [67], which reported that MSR had the best correlation with LAI [30] and yield prediction. Although the NDVI shows the greenness of the vegetation, the results show that it is the weakest predictor of rubber yield (Table 2). This is contrary to the report in [60] that NDVI combined with NIR can predict the yield of rubber well. For the mixed model, the results showed that the combination of red-edge reflectance and MSR gave the most accurate yield prediction (lowest RMSE; see Table 3).

The age range of the rubber trees in the study area was 5 to 25 years, and the average was around 12-18 years. The model developed in this study had a high yield error in 5–8-year-old para rubber [68]. Detection is difficult because it is quite difficult to distinguish these particular trees from empty areas or vegetation cover, since the trees have fewer leaves and cover only a small part of the total planted area [69]. This problem should be considered in future studies. The red edge-based method demonstrated higher sensitivity in detecting phenological and landscape changes in foliage coloration and vegetation recovery of rubber plantations [68, 70]. Although there have been some research studies on the prediction of rubber yield, there have been no studies in smallholder agriculture areas, and most of the previous studies have been done using satellite images with low spatial resolution, such as those from the MODIS satellite (with a 250-meter spatial resolution) [59]. This study was conducted with the rubber species RRIM 600. This study was conducted with the rubber species RRIM 600. Field data were collected from 213 smallholder farmers, and rubber plantations are located only in the northeast of Thailand. There may be different environmental factors with others that make for different results. Therefore, in future work, it may be necessary to test in other smallholder areas. Although the results of the prediction of the rubber yield are acceptable and higher than previously reported, there is still an opportunity to improve the accuracy. The use of reflectance and vegetation indices derived from hyper-spectral satellite data in combination with meteorological data and the application of complex mathematical models such as machine learning and deep learning is interesting for future studies.

Round 2

Reviewer 3 Report

Now the authors addressed all the comments. Accept the manuscript in present form